# The Purposes of Intellectual Assessment in Early Childhood Education: An Analysis of Chilean Regulations

**DOI:** 10.3390/jintelligence11070134

**Published:** 2023-07-06

**Authors:** Alejandro Ancapichún, Tatiana López-Jiménez

**Affiliations:** Escuela de Pedagogía, Pontificia Universidad Católica de Valparaíso, Viña del Mar 2520000, Chile; tatiana.lopez@pucv.cl

**Keywords:** intellectual assessment, assessment purposes, assessment in early childhood education, intellectual disability

## Abstract

The main purpose of intellectual assessment in early childhood education is the early detection of intellectual disabilities. In Chile, the recent school integration policy has incorporated assessment purposes oriented toward educational improvement, but it is not known how these purposes interact with each other. This study aimed to analyze the purposes of intellectual assessment present in the current Chilean educational policy in early childhood education. A systematic review of ministerial documents and a subsequent qualitative content analysis of official documents published between 1998 and 2022 were carried out. The results revealed that the purposes of intellectual assessment for educational policy are multiple, highlighting the provision eligibility, diagnosis, student monitoring and support, in addition to formative and curricular adjustment purposes. It is discussed how this multiplicity of purposes is congruent with the current regulations governing intellectual assessment procedures in early childhood education. It is concluded that there is a need to update the legislation that regulates intellectual assessment to be consistent with the new assessment proposals in the country.

## 1. Introduction

The revised literature illustrates that intelligence tests have a history of questioning ([66]), controversy, malpractice and even prosecution that have at times inhibited development and research on intellectual assessment ([30]). Despite this, intellectual assessment in educational contexts continues to have a preponderant role in the identification of intellectual disabilities in students ([39]; [59]) and contributes to educators adapting to the instruction and learning process, taking into account the student’s cognitive strengths and weaknesses ([56]).

In Chile, there is a historical relationship between the measurement of intelligence and education ([25]), which has been reflected to this day in its diagnostic role in intellectual disability. [13] ([13]) is the legal instrument that regulates the procedures for intellectual assessment. In the transition levels of early childhood education (attended by children between the ages of 4 and 6), it establishes as its primary function the identification of disorders such as intellectual deficit and global developmental delay through the application of standardized tests for obtaining a global IQ score.

In recent years, the Chilean Ministry of Education (Ministerio de Educación [MINEDUC]) has published decrees (e.g., [14]) and official orientations that promote the improvement and adequacy of instruction given the full inclusion of students with disabilities, re-orienting the uses and purposes of assessments in special education with a particular focus on early childhood education ([41]). These new frameworks advocate the formative use of any type of assessment, emphasizing that children between 4 and 6 years of age are in the process of development, discouraging diagnostic judgments.

Finally, assessment experts (e.g., [4]; [26]; [31]) agree on the importance of studying the purposes that change and are added over time in assessment systems to weigh the validity and interaction of new and old proposed uses. Otherwise, an assessment such as the intellectual assessment may serve unintended purposes under misleading assumptions. Thus, the objective of this study was to analyze the purposes of intellectual assessment suggested for early childhood education through a systematic review of official documents emanating from the Chilean educational policy published between 1998 and 2022.

### 1.1. Purposes and Multi-Purposes of Assessments in Educational Contexts

The literature points out that one of the main components of any assessment system is its purpose ([52]; [60]). In this regard, [24] ([24]) identifies two major paradigms: psychometrics and assessment for decision-making. In the former, the focus is on the design and validation of instruments for anticipated uses and purposes ([5]). The second emphasizes decisions made based on results in the context of learning and instruction ([28]).

In this way, the concept of assessment is rethought as any process of gathering information about student performance that contributes to better decision-making ([24]). For Newton, the purpose of an assessment is defined as: “which concerns the use of an assessment judgment, the decision, action or process which it enables” ([52]). Furthermore, [53] ([53]) provides an open list of purposes that affect students in the assessment process: monitoring, certification, formative, screening, diagnostic, eligibility, segregation, and placement.

When an assessment system provides results that are used for purposes that were not anticipated, the legitimacy of the decisions taken is questioned because their application is not supported by previous evidence ([54]). The literature further reports that this phenomenon is a frequent practice since, in assessment systems, the purposes change over time ([26]; [31]). The problem with this is that the administrative purposes (e.g., certification) start to take priority over the purposes oriented towards educational improvement (e.g., formative), causing the initial purposes to be misrepresented and the assessments to be carried out only to comply with administrative obligations, contributing to misguided assessment practices that can be detrimental to the students ([64]). In this way, the multipurpose phenomenon calls for periodic reviews of systems where assessments are systematically applied ([26]) and the making of adjustments to keep the initial purposes untainted ([4]), preventing the assessments from impairing students (e.g., [19]).

### 1.2. Purposes of Intellectual Assessments in Early Childhood Education

The literature on intellectual assessment (e.g., [30]; [20]; [22]) agrees that its purpose in school contexts is to identify cognitive strengths and weaknesses in students to promote better educational treatment. [18] ([18]), for example, highlight the predictive power that intelligence tests have on future academic performance, allowing the early identification of students prior to failure. [56] ([56]) point out that the results of an intellectual assessment allow the implementation of accommodations in the instructional process. Despite this, the identification of intellectual disabilities continues to be one of the most predominant purposes ([39]).

Identifying intellectual disability requires knowing a student’s intellectual functioning based on the measurement of a set of cognitive abilities that are represented in an overall IQ score ([16]). For this task, psychologists resort to the application of objective cognitive test batteries. This intellectual assessment contributes not only to identifying people with intellectual disabilities who have significantly below-average performance but also to the development of support plans aimed at improving the interaction between the person and their environment ([59]).

Other purposes of intellectual assessment include the identification of gifted students, vocational guidance according to a profile of cognitive strengths and weaknesses ([56]), and determining eligibility to access special education services ([29]). This depends on the fulfillment of certain criteria presented in the legislation of each country. The IQ 70–75 cut-off score is one of the criteria presented in both Chile and the USA that guarantees the access of students with intellectual disabilities to special education services ([7]). 

In early childhood education, the practice of intellectual assessment is strained by the dilemma of identifying developmental disabilities and early impairments ([16]). On the one hand, the risk of a diagnosis devolving into a label that limits developmental opportunities is increased by the reliance on standardized tests that may be misused and inappropriate in young children ([50]). For example, the IQ scores obtained by this population are not very stable over time ([27]), and many testing procedures are designed for older children, increasing the risk of misdiagnosis ([21]). Because of this, the use of fixed cut-off scores for early identification of intellectual disability is problematic ([16]; [33]).

On the other hand, early identification and intervention predict better levels of educational achievement and social adjustment, with standardized tests also being tools that help educators make better developmental and instructional decisions ([6]). In countries such as New Zealand ([2]) and the USA ([37]), screening and referral are hampered by the belief that the child will grow up, limiting the developmental opportunities of children with disabilities. 

Organizations that address child development, such as the [15] ([15]) and the [51] ([51]), propose guidelines for Young Child assessments, such as considering the child’s developmental environment ([22]) and involving the family in the process, with less emphasis on the application of tests. These guidelines are incorporated into other assessment approaches such as the Authentic Assessment ([3]), Response-to-intervention ([62]) and Collaborative–Adaptive Student-Centered CASC ([61]) frameworks. In this way, children can benefit from an early diagnosis that does not harm their developmental process, which is the intellectual assessment key to the recognition of intellectual disability ([16]).

### 1.3. Educational Inclusion Policy in Chile

The pillars of the current Chilean educational system are access, inclusion, and equal opportunities for all students, especially those who present special educational needs (SEN). The Chilean regulatory framework defines students with SEN as those who require extra support and resources to access, remain in and progress in the educational system. The concept of SEN includes students with intellectual disabilities, autism, specific learning disabilities and other developmental conditions that interact with environmental barriers that hinder their learning process. For a student with SEN to access these special supports and resources, they must undergo different types of assessment according to the diagnostic hypothesis and established procedures. Intellectual assessment is a type of SEN assessment that is mainly performed in cases of intellectual disability, global developmental delay and specific learning disabilities ([13]).

In the last two decades, Chilean educational policy has taken important steps towards the integration of children and adolescents with disabilities into regular classrooms and establishments through the so-called School Integration Programs (PIE in Spanish). PIEs are agencies responsible for identifying students with SEN through a team of professionals from early childhood to high school levels for their subsequent inclusion in regular classrooms. In turn, the MINEDUC Studies Center points out a significant increase in professionals who perform this type of assessment in schools, from 7.899 in 2009 to 45.042 in 2018 ([8]). These figures evidence that the number of students assessed by these professionals and integrated into general establishments has increased significantly, in tune with the proliferation of these programs in the country. This shows that in the last 10 years, SEN assessments, and more specifically, intellectual assessments, have become a phenomenon that has spread on a national level. However, the implementation of this policy in schools presents at least three problematic dimensions linked to the assessment process.

The first dimension refers to the fact that schools subscribed to the program fail to improve, diversify, and adapt the instruction (e.g., [36]). Empirical studies mention practices and social effects that go in the opposite direction of the desired social inclusion of children in regular schools (e.g., [49]; [55]), which is attributed to the role of the assessment process in PIE. The literature highlights that assessment is an obstacle because professionals dedicate greater efforts to obtain diagnoses to the detriment of accompanying teachers in making educational improvements in their instruction practices (e.g., [1]; [35]). [65] ([65]) express that PIE professionals prioritize the assessment process, dedicating more time to it than to the process of improving and adapting instruction. On the other hand, [9] ([9]) and [10] ([10]) emphasize that the assessment in PIE helps schools visualize the needs of students who were previously excluded from the general education system and thus determine educational adjustments for their benefit.

In addition to the specific problems of the Chilean reality, there is the fact that the literature reports mixed results on the full inclusion of students with SEN ([9]; [57]). The studies reveal improvements in social competence ([17]); however, there is insufficient evidence of solid and sustained academic progress (e.g., [63]).

The second problematic dimension is the financing of the PIE, which is based on an additional subsidy given to the schools per diagnosed child. These additional resources are established by legal regulations and allow the PIE to function year after year. However, some experts (e.g., [23]) criticize this form of funding because it could encourage overdiagnosis of students with SEN. Regarding this aspect, empirical evidence is scarce, but rates of overdiagnosis of SEN associated with attention-deficit/hyperactivity disorder have been observed ([58]).

The final problematic dimension is seen in the assessments of early childhood education levels to enter PIE. The recent curricular reform prioritizes full inclusion as a pedagogical principle and the comprehensive development of children, respecting their characteristics ([46]). This leaves little to no opportunity for the application of any type of assessment that does not have formative purposes. Thus, intellectual assessments that have diagnostic purposes are in contradiction with these principles, which could discourage the early identification of intellectual disabilities.

Experts (e.g., [38]) point out the need for school integration policies to move from a paradigm of SEN classification to a paradigm of inclusion, where individual diagnosis has a less preponderant role. From the Ministry of Education, attempts have been made to remedy the problems that originate from the assessment in the PIE by adding evaluative purposes aimed at improving and adapting instruction (e.g., [14]). Therefore, if the purpose of intellectual assessment in early childhood education before was only the diagnosis of intellectual disability, today it must satisfy other additional purposes due to this gradual change in school integration policy. Currently, a knowledge gap exists around identifying what these purposes are and how they interact with each other.

### 1.4. The Present Study

The objective of this study was to analyze the purposes of intellectual assessment proposed for early childhood education through a systematic review of Chilean educational policy. A qualitative content analysis of official documents published between 1998 and 2022 was carried out, contrasting the purposes of intellectual assessment with the current regulations ([13]), to develop recommendations for the updating of these regulations.

## 2. Materials and Methods

The qualitative design of this study enabled the analysis of meanings present in documents emanating from Chilean educational policy ([40]). The methodological approach is similar to that used by [34] ([34]), in which the normative documents were identified, then a selection was made, and a subsequent qualitative analysis of these documents was carried out. The guide questions were as follows:

What purposes of the intellectual assessment are stated in the technical and guidance documents?

Are the purposes identified in the technical and guidance documents consistent with current regulations?

### 2.1. Identification and Selection of Normative Documents

In Chile, the Ministry of Education is the agency responsible for leading educational policy through the promulgation of decrees that establish the legal framework of action for schools, as well as the creation and publication of technical and guidance documents. This research focuses on the review and analysis of technical and guidance documents related to [12] ([12]) and [13] ([13]), which establish and regulate the procedures for intellectual assessment in early childhood education. The search and selection of the documents were carried out in the official database of special education of the Ministry of Education (https://especial.mineduc.cl/ (accessed on 15 February 2023)), in which all official documents referring to special education are listed. Because of this, inclusion and exclusion criteria were established.

The first inclusion criterion was that the guidance and technical document had as a reference [12] ([12]) or [13] ([13]). A second inclusion criterion was that the document stipulated its application to the early childhood education level. In order to answer the questions proposed in this study, documents such as instruction guides, which address classroom intervention for specific SEN, were excluded from the analysis. Documents such as Frequently Asked Questions and research reports were also excluded.

### 2.2. Textual Corpus Obtained

From the database search, a total of 52 documents were found, to which inclusion and exclusion criteria were applied. In a first review, 7 documents published in 2008 were excluded because they dealt with the assessment process in early childhood education by type of diagnosis without mentioning the decrees. Subsequently, 18 pedagogical guides and 7 research reports were excluded. Then, 13 documents were excluded because they dealt with educational levels other than early childhood education. Finally, a total of 7 guidance and technical documents were included (see Table 1) that the Ministry of Education currently makes available to orient intellectual assessment practices in early childhood education.

It is important to note that the seven documents analyzed use the term “SEN assessment,” which incorporates intellectual assessment and therefore applies it. Also, all the guidance documents refer to [13] ([13]) and not to [12] ([12]).

### 2.3. Data Analysis

The analysis of the technical and guidance documents consisted of a qualitative content analysis, which reduced the data into descriptive categories that provided answers to the research questions ([11]). Such analysis was supported by the software ATLAS.ti version 23. The analysis process was divided into three stages: first, a codebook was developed; second, a thematic reduction was performed; and finally, the categories were contrasted in light of the current regulations ([13]).

First, codebook development was developed deductively from the eight assessment purposes proposed by [52] ([52], [53]) that refer to the student level. The unit of analysis was of a thematic type, coding segments corresponding to one or several sentences present in a paragraph in an excluding manner, according to rules associated with the definition. Table 2 presents the assessment purposes, an operational definition, and their respective coding rules. To ensure the reliability of the analysis process, both authors previously agreed, reviewed the coding rules and made a parallel analysis of each of the documents, obtaining a high level of agreement in the coding of the identified segments. When the identified loose segments could not be classified in the previous domains, the elaboration of a new code was discussed and consensually agreed upon.

Once all the text was reviewed and coded, we proceeded to redefine each purpose based on its coded segments, obtaining rich descriptions of each assessment purpose in the terms proposed by the documents. Then, the identified assessment purposes with their respective descriptions were grouped into categories ([11]) according to the theoretical criterion of assessment function. Thus, it was determined that the identified purposes could be grouped into two categories, one referring to the function of educational improvement and the other to the function of certification and classification of the special condition (see Figure 1). Finally, these two categories that resulted from the analysis of the technical and guidance documents were contrasted with the assessment approaches stipulated in the regulations for early childhood education ([13]). The contrast considered two dimensions of analysis referring to the degree of consistency between the regulations and their purposes, leaving on the one hand the evaluative procedures and, on the other, the purposes.

## 3. Results

The results are organized into two sections that respond to the research questions proposed in this study. On the one hand, the purposes identified from the analysis of the technical and guidance documents are described. On the other hand, the consistency of these purposes in relation to the current regulations is addressed.

### 3.1. Assessment Purposes

From the analysis of the seven technical and guidance documents, four of the eight assessment purposes proposed by [53] ([53]) were recorded, namely: provision eligibility, diagnosis, student monitoring, and formative. In addition, two other purposes that emerged from the data were identified, namely curricular adaptations and supports. Curricular adaptations refer to pedagogical decisions aimed at adapting instruction and materials within the classroom according to individual assessment results, while supports involve psychoeducational intervention actions (e.g., individual treatments) executed by PIE professionals (e.g., a psychologist or a special educator) inside or outside the classroom. Therefore, the first result shows that, for Chilean public policy, intellectual assessment in early childhood education is configured around these six assessment purposes.

Additionally, to facilitate the characterization of similarities and differences between the assessment purposes identified, they were grouped into two continuous categories: educational improvement on the one hand, and certification and classification of the special condition on the other (see Figure 2). Each category and its respective assessment purposes are described below.

#### 3.1.1. Assessing to Certify and Classify the Special Condition

This category gathers the purposes of assessment, whose aim is to certify the child’s special condition by means of an act of classification and labeling that is supported by professional and/or legislative criteria. It integrates two purposes: provision eligibility and diagnosis. The eligibility purpose refers to the fact that the intellectual assessment is intended to determine whether the child being tested meets the criteria for entering the PIE. This is presented in documents B and E, which guide the opening, administration, and technical operation of the PIE, stating that the assessment “constitutes a legal requirement to allocate the resources allocated by the state to the special education subsidy for school integration programs (PIE)” (Document E, p. 23).

However, the documents that specifically guide the early childhood education level (A, D and F) do not mention this purpose. A possible reason for this omission is made explicit by the most recently published document (G), which orients the PIE assessment professionals in the following way:

In many cases, this role [eligibility] has tended to be visualized and/or assumed in a bureaucratic manner, with the sole purpose of validating, administratively, that the student presents the deficit, disorder or disability condition that will allow him/her to be incorporated into the PIE and receive the special education subsidy.(Document G, p. 16)

According to document G, the assessment that has the purpose of determining eligibility misses other applications of the results to educational improvement since the focus is on establishing whether the evaluated child meets the legal requirements for formal admission to PIE stipulated by [13] ([13]). One of these requirements, relevant in the case of the intellectual assessment, is the presence of an IQ below 70 points, which classifies the child under the diagnosis of global developmental delay or intellectual disability.

The second purpose considered in this category is that of diagnosis, which refers to identifying and describing the SEN in a child independent of their formal entrance to the PIE. This purpose is mentioned in documents A, B, D, E and G. For example, document A, which is the oldest document aimed exclusively at the early childhood education level, describes the purpose of the assessment process as follows:

Its purpose is to identify, broadly speaking, those SEN that require primary attention due to their impact on the student’s development and learning process, and that also imply extra measures of a temporary or permanent nature.(Document A, p. 46)

Nevertheless, documents A, B, D, E and G conceptualize SEN in different ways: either as impairments (E), learning support needs (E, G), disabilities (E), disorders (E), or difficulties that affect the learning process (A, E, G). Subsequently, these documents clarify that SENs are conditions that derive from a disorder, disability or impairment and do not constitute individual attributes of the child. For this reason, documents such as A suggest that the assessment should address not only individual dimensions of the child (strengths and weaknesses) but also contextual dimensions in order to understand the variables that affect SEN through the use of criteria specific to each discipline:

The assessment (…) focuses not only on the determination of the student’s difficulties but also on their potential, as well as on the identification of all those factors of the educational and familial context that may influence their educational progress.(Document A, p. 16)

It could be argued that the purpose of intellectual assessment in children in early childhood education is not reduced to identifying a disability, such as an intellectual deficit. It also implies understanding the learning condition that derives from the interaction between the child and their environment and responding through educational actions for much better treatment. The above is strengthened by what is stated in documents B and E on the importance of guaranteeing the educational rights derived from the presence of a special condition: 

If the diagnostic assessment determines the presence of SEN, the student should be subjected to psychoeducational intervention, for which, if they meet the criteria indicated in the regulations, the establishment can apply to a School Integration Program (PIE). If they do not meet them, other measures must be taken to support their learning process.(Document B, p. 24)

In summary, both the purpose of diagnosis and that of eligibility imply classifying the student according to certain criteria to accredit their special condition. In the former, professional judgment prevails, while in the latter, the criterion of legal regulations predominates. In the former, the assessment has the consequence of guaranteeing the rights of the child’s special condition to the entire educational system, while in the latter, its only consequence is the child’s formal admission to the PIE. According to the emphasis placed on classification and labeling, eligibility is closer to the left end of classification and certification of the special condition, while diagnostic is farther away because it lacks explicit legal criteria (see Figure 1).

#### 3.1.2. Assessing to Improve Educational Processes

This category brings together the purposes of assessment that aim to improve instruction and learning processes within educational establishments through different actions that may occur inside or outside the classroom. The purposes alluded to are the following: support, curricular adaptations, student monitoring and formative. The purpose of the support is to promote educational improvement through the planning of special assistance for children with SEN. This is present in the seven documents analyzed, which coincide with defining support as pedagogical, material, and/or human resources complementary to those usually provided by the classroom teacher. These educational supports are aimed at reducing or eliminating the barriers to learning, development and participation faced by children with SEN.

The first technical document of the 2013 PIE (B) specifies that supports are psychoeducational interventions of an individual nature implemented inside or outside the classroom by PIE professionals, whereas the specific documents for early education (A and F) advocate for environmental interventions that modify the school context, benefiting the entire educational community. Lastly, document G, when referring to the functions of the professional evaluators of the PIE, points out that a central task of the evaluator is the following: “To report the conclusions of the comprehensive assessment and identify the central support actions for the student in the area of the specialty or professional discipline” (Document G, p. 17). It is deduced, therefore, that the type of support (individual or environmental) will vary according to the specialty and focus of the assessing psychologist.

Regarding the purpose of curricular adaptations, which refers to the determination of curricular modifications, this is present in four of the seven documents (B, C, D, and G) and is recurrently mentioned in the documents that guide the diversification of instruction within the educational system (C and D). In both documents, curricular adaptations are defined as a type of pedagogical support that involves a set of actions carried out by teachers in the classroom to support learning in young children’s education. That is, modifications aimed at implementing the curriculum in the classroom considering the individual differences of students with SEN in order to ensure their permanence, progress and participation in the school system. Finally, these actions are decided between the special educator of the PIE and the classroom teacher according to the results of the assessment (Document C, p. 36).

The purpose of monitoring, also included in this category, is to know how children are progressing in their learning experience and to determine how relevant and effective the implemented supports are. It is mentioned in documents B, F, and G, giving it a secondary role compared to the other assessment purposes (Document G). But it is mandatory for children with disabilities who are formally part of a PIE (Document B). This is because the information provided by the assessment results contributes to modifying or maintaining the supports implemented.

Finally, the last purpose present in this category is formative. It refers to the pedagogical actions designed to improve the instruction and learning that occur inside the classroom, benefiting all students. This purpose is relatively new, being mentioned exclusively in the last two published documents (F and G). It is stated that the results of the assessment should be useful for the classroom teacher, allowing them to reflect with their pedagogical team and thus improve their instruction: “Assessment and monitoring should be a permanent, constant process since it has to provide relevant inputs to improve strategies that promote pertinent learning in all children” (Document F, p. 22). Indeed, the decisions made do not exclusively benefit children with SEN, but all the children in the classroom. Any type of assessment, including intellectual assessment, should be conducive to improving the instruction and learning of all students as a collective: “Psychoeducational assessment should serve to guide the educational process as a whole, facilitating the task of the teacher who works with the student daily” (Document G, p. 17). Hence, the role of the early childhood educator in this decision-making process is fundamental.

In summary, these four purposes that make up the category “assessing to improve educational processes”, promote actions that seek to encourage educational actions in support of the instruction and learning process. Depending on the pedagogical focus of these actions, some will be closer to the right end of the educational improvement spectrum, and others will be farther away from this extreme (see Figure 1).

### 3.2. Consistency of the Purposes Identified with the Current Regulations 

The second result indicates the consistency between the purposes identified in the technical and guidance documents and the current regulations governing intellectual assessment procedures. For this, two dimensions of analysis were established: the viability of these purposes in relation to the assessment procedures stipulated in the regulations and the degree of coincidence between the purposes identified in relation to the original purpose.

#### 3.2.1. Viability of the Identified Purposes in Relation to the Current Regulations

This analysis dimension refers to whether the regulations state and mandate clear, pertinent and specific assessment procedures to satisfy each purpose identified in the technical and guidance documents. The educational regulation ([13]) declares and stipulates a single valid procedure for intellectual assessment, consisting of the application of standardized cognitive tests to obtain a global IQ score: “Significant limitations of intellectual functioning are expressed with a score equal to or less than 69 points on a standardized intelligence test” (p. 16). However, it is clarified that only in exceptional or atypical cases may the psychologist follow alternative procedures such as clinical observation, play-based assessments, graphic tests and scales or questionnaires that are not standardized in Chile. In both cases, the final product of the assessment process must be a summative judgment on the level of intellectual functioning of the child being assessed (e.g., borderline, mild intellectual deficit): “The clinical judgment of the specialist will be used to determine the degree of limitation of intellectual functioning” ([13]).

The application of intelligence tests to obtain IQ scores and the use of clinical judgment present in the current regulations make it viable to fulfill the purposes of certification and classification of a child’s special condition in the form of eligibility (Documents B and E). As the regulations state, the results of the intellectual assessment must categorize a child as having a type of intellectual disability. This classification is obtained by obtaining an IQ below 70 points, which determines a child’s eligibility for special services from a PIE (Document E). However, this procedure of intellectual assessment focused on IQ (Decree 170 of 2009) does not allow the proper achievement of the diagnostic purpose of SEN, which is to understand the educational needs that derive from the child’s interaction with their environment as established in the documents specified for early childhood education (Documents A, D and F).

In line with the above, the procedures stipulated in the regulations make it impossible to obtain useful and relevant data for the improvement of educational processes from an intellectual assessment in the manners proposed in the technical and guidance documents (A, B, C, D, E, F and G). Indeed, the regulations do not establish other assessment procedures that comply with this set of purposes associated with educational improvement.

#### 3.2.2. Coincidences between the Document Purposes and Current Regulations 

The second analysis dimension refers to the coincidence that exists between the six purposes and the original purpose stipulated in the regulations. In other words, it implies determining the content consistency between each purpose proposed by the technical and guidance documents in relation to the original purpose currently mandated for intellectual assessment in the Chilean educational system.

The regulations ([13]) state two purposes that the intellectual assessment must fulfill: “(…) identifying the intellectual disability and determining the type of supports that should be provided to the student” (p. 16). Both purposes, according to the regulations, are fulfilled with the incorporation of the results of other assessments (e.g., adaptive behavior assessment), all of which lead to the admission of the student to the PIE. Both purposes of intellectual assessment in the Chilean educational system are consistent within only two of the six purposes suggested in the technical and guidance documents, namely, eligibility (Documents B and E) and support (Documents B, C, D and G). Consequently, the technical and guidance documents maintain the two original purposes present in the regulations, clarifying their scopes and consequences. At the same time, they add four new purposes (diagnosis, monitoring, curricular adaptations, and formative) that are not present in the current regulations ([13]).

In the case of intellectual assessment in children between 4 and 6 years of age, the coexistence of the two original purposes (eligibility and establishment of support) and four additional purposes (formative, diagnosis, curricular adaptations, and monitoring) is evidence of the multipurpose assessment phenomenon. The four additional purposes do not respond to the purpose that the Chilean educational policy initially proposed for intellectual assessment when it designed the system in the PIE. The addition of these four purposes responds to the changes made in the last 10 years to the assessment policy in early childhood education, which are manifested in the technical and guidance documents. Thus, these documents encourage applications that differ from the original purposes.

In synthesis, the current regulations ([13]) are consistent with the eligibility purpose suggested by the technical and guidance documents, both in the procedures and in the mandated assessment purposes. Likewise, this regulation is scarcely consistent with the educational improvement-oriented purposes presented in the technical and guidance documents.

## 4. Discussion

Assessment purposes are at the core of every evaluation system. The findings of this study point out the presence of six purposes in the technical and guidance documents. These findings are discussed here in the context of the current regulations. In addition, the interaction between these multi-purposes is projected for the case of intellectual assessment in children from 4 to 6 years of age.

Both the current regulations and their guidance documents consider that intellectual assessment has a key role in the identification of intellectual disability in children, which is consistent with the recent guidelines of the American Association on Intellectual and Developmental Disabilities ([59]). In consequence, obtaining an IQ score and/or a judgment about a child’s intellectual functioning as a result of the assessment allows the fulfillment of one of the criteria for intellectual disability ([16]). The above is a relevant legal aspect that guarantees the access of the child with a disability to education services, which is consistent with the purpose of eligibility for special services observed in regulations in other countries such as the USA ([32]).

From the analysis of the guidance documents, it can be inferred that, for the current Chilean educational policy, the intellectual assessment also contributes to the diagnostic purpose of SEN, going beyond the identification of an intellectual disability. The SEN diagnosis aims to understand the educational needs of a child from a contextual point of view that considers individual and environmental dimensions. This broader understanding of diagnosis coincides with [52]’s ([52], [53]) definition, which defines it as the clarification of the extent and type of difficulties that a student presents, thus differentiating it from the purpose of eligibility. Therefore, the assessment process should not be reduced solely and exclusively to the application of standardized tests to obtain IQ scores. Instead, it should incorporate other approaches that are more conducive, ecological and consistent with children’s development ([62]), such as authentic assessment ([3]) or the Collaborative–Adaptive Student-Centered (CASC; [61]) assessment.

The above suggests that the diagnostic purpose is closely linked to other purposes, such as the determination of curricular supports and adjustments. For example, the analysis of a cognitive profile of strengths and weaknesses ([56]) or the incorporation of valuable information related to the family context can provide useful results both for the diagnosis of SEN and for instruction and curricular adaptation ([30]).

On the other hand, there is no theoretical support in the specialized literature regarding the formative purpose that the most recent guidance documents propose for intellectual assessment. For [53] ([53]), this purpose allows the identification of learning gaps that guide and improve instruction and subsequent student learning. However, in the case of intellectual assessment, there is no overlap between what is assessed and what is taught by a teacher in the classroom. Indeed, intellectual assessment measures aptitude while educators develop skills ([62]), making it difficult, if not impossible, to understand its results pedagogically. Likewise, teachers in early childhood education have a referential framework for a curriculum that considers the development of skills and abilities to be linked to highly specific contents and contexts, making the usage of cognitive assessment results to guide learning objectives and goals nonsensical ([20]). 

The results manifest the expectation that Chilean educational policy places on intellectual assessment in early childhood education; in addition to certifying and classifying the special condition of a young child, it should also contribute to four additional purposes of educational improvement. From the point of view of the multi-purposes of an assessment, this density of purposes is problematic ([53]), especially when they coexist with mandatory administrative purposes. This complements [64]’s ([64]) statement that administrative purposes detract from others aimed at educational improvement because they acquire greater importance to the educational system.

These results also show that Chilean legal regulations establish mandatory intellectual assessment procedures that ensure access to economic benefits for institutions for every child identified as having an intellectual disability. Theoretically, the purpose of eligibility acquires supremacy over all other purposes when the assessment is carried out to comply with these legal requirements to the detriment of improving and adapting instruction. This supremacy may have additional negative consequences if we consider that one of the criteria for determining the eligibility of a child with an intellectual disability for special education services (i.e., PIE) is a fixed one.

Although a cut-off score of 70 IQ points obtained in an intelligence test provides objective information for decision-making, it also sets a barrier associated with the scarcity of available tests for the 4–6 age group in Chile. Likewise, the moderate instability of IQ scores ([27]) does not guarantee an early identification of intellectual disability with the exclusive use of tests. To circumvent this barrier, PIE psychologists could introduce forms of assessment focused on accompanying the children’s development, attending to evolutional milestones, and recognizing warning signs. This assessment approach addresses the rapid changes that children undergo in their development without fragmenting the focus toward fixed and isolated scores. It is essential to promote forms of assessment that use flexible criteria for children to ensure access to special aids for children with suspected intellectual disabilities or developmental delays ([7]). Although in Chile the literature has questioned the PIE program’s assessments (e.g., [1]), this criticism should focus on the eligibility purpose that has distorted the functioning of these programs, making it necessary to reevaluate the benefits of the assessment usage and purposes ([9]; [10]) oriented to the improvement suggested by the documents.

### Study Limitations and Future Research

This study focused on normative documents that defined and framed the purposes of intellectual assessment in early childhood education in the Chilean assessment system. Firstly, although the adopted design allows for describing the proposed assessment purposes, it is not possible to deepen the understanding of these purposes by the subjects involved or to characterize research or intellectual assessment practices in early childhood education. Secondly, the findings of this study can only be extrapolated to educational systems where intellectual assessment plays a predominant role in the identification of intellectual disability. And finally, regarding the identification of multi-purposes, it would be convenient to continue investigating to know the representations of the educational actors and to explain the critical ties from both points of view. Future research could be opened in the line of intellectual assessment practices to characterize how they are carried out, under what procedures, and how they satisfy all the identified purposes, as well as explore the situational representations that the actors hold about the multi-purposes of intellectual assessment.

## 5. Conclusions

Initially, this study identified six purposes that the Chilean educational policy attributes to intellectual assessments in early education: formative, establishment of support, curricular adaptations, monitoring, diagnosis and eligibility. These six purposes can be grouped into two categories: on the one hand, the certification and classification of the special condition, and on the other, the educational improvement. Out of these purposes, only the purpose of eligibility is consistent and viable to meet the current regulations in Chile, taking precedence over the other purposes of educational improvement.

The aforementioned reveals the tension that exists in educational policies regarding multi-purposes. To overcome this situation, three recommendations are presented to update the Chilean regulations and make them consistent with the multiple purposes identified for intellectual assessment in early childhood education.

Firstly, regarding the criteria of eligibility for special education support, it is expected that these are consistent with the definition of SEN in our educational system. An IQ score below 70 points as well as the clinical judgment of deficient intellectual functioning are not relevant as eligibility criteria for the initial education level. This is because SEN is understood as an interaction between the child and the environmental barriers, and it is the task of the intellectual assessment to identify these complex ties from the cognitive skills’ viewpoint. 

Secondly, in line with the assessment procedures for children in early childhood education, the regulations should stipulate relevant assessment approaches that, for example, incorporate the family and educators in the process. This is with the intention of satisfying the purposes oriented to educational improvement and the opportune development of young children with disabilities in full-inclusion educational contexts.

Finally, regarding the derivation of a child from early childhood education to intellectual assessment, the regulations should incorporate mechanisms that ensure the fulfillment of the purposes oriented toward educational improvement. This prevents the problem of multi-purposes.

## Figures and Tables

**Figure 1 jintelligence-11-00134-f001:**
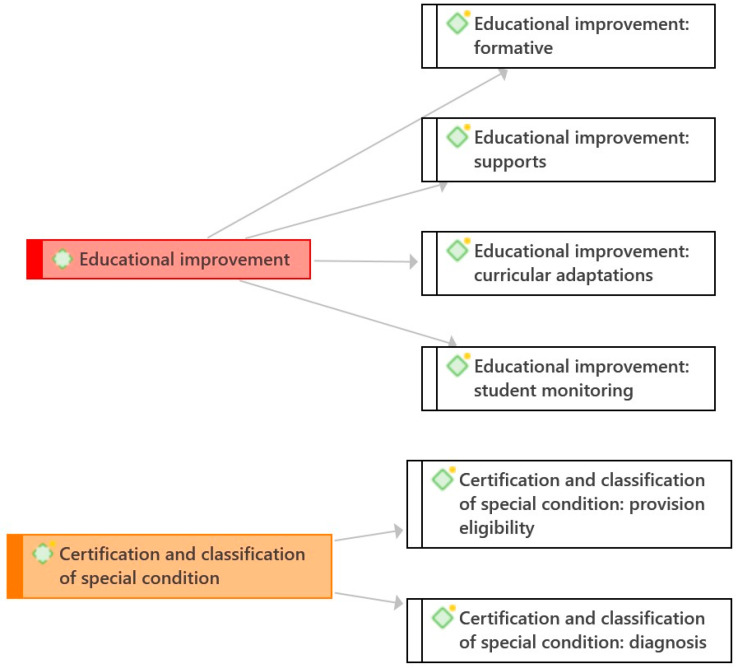
Categories and grouped codes.

**Figure 2 jintelligence-11-00134-f002:**
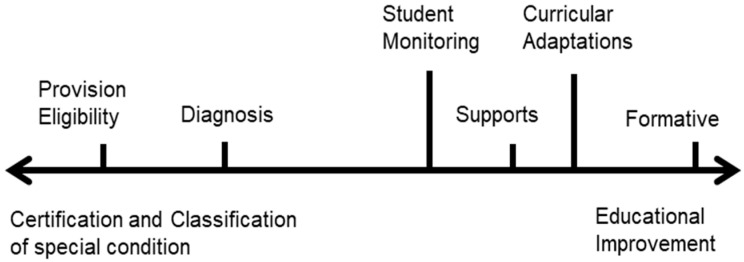
Identified intellectual assessment purposes.

**Table 1 jintelligence-11-00134-t001:** Selected technical and guidance documents.

Alphabetical Identifier	Document Name	Target Level
A	Guidelines for educational responses to diversity and special educational needs (Ministerio de Educación [41])	Early childhood education
B	Technical guidelines for school integration programs ([42])	All educational levels
C	Criteria and guidelines for curricular adaptation for students with special educational needs in early childhood education and elementary education ([43])	All educational levels
D	Guidelines on diversified instruction strategies for elementary education in the framework of decree 83/2015 ([45])	Early childhood education
E	Support manual for the implementation of PIE ([44])	All educational levels
F	Guidance document for the development of inclusive practices in early childhood education ([47])	Early childhood education
G	Guidance on the role and functions of education assistant professionals who participate in school integration programs (PIE) ([48])	All educational levels

**Table 2 jintelligence-11-00134-t002:** Definition of preliminary codes for assessment purposes.

Term	Definition	Codification Rules
Student Monitoring	Determine whether students are progressing adequately over time against specific learning or intervention objectives.	Rule 1. This applies when the purpose is to determine the adequacy of supports and interventions while they are being provided. Rule 2. Does not imply an improvement in the classroom teacher’s instruction.
Learning Certifications	Indicate whether students have met the requirements (knowledge, skills, etc.) of a given course.	This applies when it is intended to indicate whether students have met the requirements associated with the knowledge, objectives, and skills of the level.
Formative	Identify student learning gaps or needs to guide and improve subsequent instruction and learning.	Rule 1. This applies when the classroom teacher decides to modify and improve their instruction for all the students in the class. Rule 2. This does not apply when the decision is made by the special education teacher.
Screening	Identify students who differ significantly from their peers in certain areas or dimensions to deepen assessment.	This applies when the decision involves referring the child for any kind of further evaluation.
Diagnosis	Clarify the type and extent of the student’s learning difficulties in light of specific criteria.	This applies when the decision is to establish a category, label, typology, or description that communicates the student’s learning difficulties or the extent (degree) of the difficulty.
Provision Eligibility	Determine whether the student meets the criteria for accessing special education services and aids.	Rule 1. This applies when the decision is whether to enter the School Integration Program (PIE) or not. Rule 2. This applies also when the decision is the discharge from the PIE program.
Placement	Placing students at particular instructional levels or in particular educational programs that are more educationally enriching for them.	This applies when the decision is to place students in certain educational levels or curricular acceleration programs.
Segregation	Segregate students into homogeneous groups based on ability or achievement to make instruction simpler or more viable.	This applies when the decision is to group students within the classroom or in special classrooms according to their aptitudes.

Source: based on [52] ([52], [53]).

## Data Availability

Publicly available datasets were analyzed in this study. This data can be found here: [https://especial.mineduc.cl/ (accessed on 15 February 2023)].

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
