# Peer review of "The Purposes of Intellectual Assessment in Early Childhood Education: An Analysis of Chilean Regulations"

_jintelligence, 2023, doi:10.3390/jintelligence11070134_

Round 1

Reviewer 1 Report

The article is a strong fit for the special issue, and provides much-needed and valuable information that can expand our international boundaries of our knowledge on the use of intellectual assessment. In its current form, the paper struggles to maintain its primary focus, which is to provide a qualitative examination of educational procedures and policies in Chile. The introduction and section 1.1 could probably be consolidated and shortened. In its place, an expansion of sections 1.2, and specifically the focus on SEN, would make the paper a stronger alignment with the stated intention of “early detection of intellectual disability.”

In addition, pointing out the parallels between Chile’s diagnostic criteria and the criteria of the US (or other countries) would help show the importance and wide-spread impact that these policies can have on the lives and opportunities of school children in Chile. For example, presenting the findings on the use of IQ and the 70 cut-off score for intellectual disability within the US (a diagnostic criteria that is also present in Chile), could underscore the importance of inconsistencies between the documents examined in this manuscript.

Further, a brief discussion on the implications of the “ages 4-6” for these diagnoses within the context of the reliability/validity/lifelong stability of intellectual assessments geared for these ages would also be helpful. On one hand, children are getting identified and receiving useful services at a critical stage. On the other hand, they may also be stigmatized unnecessarily at the same critical age. Is there evidence of income or other group disparities on the efficacy of SEN services? Further, there may be children who do not exhibit their symptoms until a later age, which may lead to another array of consequences.  

In addition, the information on lines 181-186, where schools are incentivized to give ADD diagnoses is eye-catching and could be explained further. What, exactly, are these incentives? Is there empirical evidence of this? What role does/could intellectual assessments play in ADD diagnoses in an educational setting and does this role change for ADD diagnoses within a clinical setting (in the US, educational diagnoses are independent of clinical diagnoses).  Also, it is arguable that this can apply to all diagnoses if schools are given extra funds for SEN with other diagnoses (for example, in the United States, schools are given more money for children diagnosed with Emotional Disturbance, Autism, and Intellectual Disability compared to Learning Disability).  

A minor recommendation: State the “original” and “additional” purposes that are alluded to in lines 541-548.  

Author Response

Dear reviewer,

We genuinely appreciate your comments and observations as they contribute to improving the focus of the information presented. The implemented improvements are detailed point-by-point as follows:

  1. In order to enhance the main focus of the study, we have reduced Section 1.1 emphasizing the idea of assessment purposes. Additionally, we expanded Section 1.2 to cover concerns regarding the stability of IQ scores, the use of fixed cut-off scores, and the utilization of standardized tests in young children. The Introduction remained unchanged as it effectively connects Section 1.1, Section 1.2, and Section 1.3, addressing the research problem seamlessly.
  2. Likewise, in the discussion section, we have elaborated two paragraphs that address the problematization of inconsistencies in ministerial documents regarding the use of fixed IQ scores for eligibility purposes, anticipating potential negative consequences.
  3. Regarding overdiagnosis in PIEs, we have restructured the ideas to properly explain the impact of incentives on overdiagnosis. We have included the only evidence from Santana-Vidal et al. 2020 regarding the ADDH case. This is a critical topic in national education policy.

We once again express our gratitude for your valuable suggestions.

Reviewer 2 Report

Dear Authors,

As a reviewer of the work, I would like to thank you for raising an important topic and for its systemic and extensive elaboration.

I would also like to ask you to consider the following changes, which do not have a significant impact on the quality of the work and systematize its structure or the quality of the presented graphic materials:

1. Materials and methods: it seems to me that I am not justified in dividing this part into smaller ones - especially since in some cases individual sections are very short

2. Figure 1: the presented graphic material should be changed in terms of its size and quality (resolution)

3 Discussion: please consider a different way to start a discussion than by saying: Below we ......

4. Conclusions: I am asking that the conclusions contain only the issues discussed earlier without citing other authors (this should be done in the appropriate part of the work).

I wish you good luck and quick publication of your work,

Reviewer

Author Response

Dear reviewer, we greatly appreciate your comments. We value and welcome all of your suggestions as they contribute to improving the way the information is presented. The applied improvements are detailed point-by-point as follows:

  1. The Materials and methods section has been reorganized into three subsections: 2.1 Identification and selection of normative documents, 2.2 Textual corpus obtained and 2.3 Data Analysis.
  2. The image has been enlarged and the pixel resolution improved.
  3. The beginning of the discussion section has been modified by mentioning the main finding of the research: “Assessment purposes are the core of every evaluation system. The findings of the study point out...”
  4. The wording of the conclusion section has been redrafted in order to remove quotes.

We sincerely appreciate your valuable suggestions.

Round 2

Reviewer 1 Report

The authors have succesfully revised the paper. There are a few times where paragraphs are edited in a manner where only 1 sentence remains, which is awkward from a grammar and stylistic view. Therefore, a final read-over and edit would be useful, but does not necessitate another review. 

Author Response

Dear Reviewer

We once again express our gratitude for your valuable suggestions.
We have edited those points. Thank you for helping us improve the paper.

Kind regards,
authors

Reviewer 2 Report

Dear Authors, 

Thank you for including the suggestions in the submitted version of the article. I wish you fruitful further scientific work and taking up equally important practical issues. 

Kind regards,
Reviewer

Author Response

Dear Reviewer, 

We appreciate your comments and desires.

Kind regards.